# An Assessment of Horse-Drawn Vehicle Incidents from U.S. News Media Reports within AgInjuryNews

**Nicole Becklinger**

Department of Engineering, University of Southern Indiana, Evansville, IN 47712, USA; nlbeckling@usi.edu

**Abstract:** Some old-order Anabaptist communities rely on animal-drawn vehicles for transportation and farm work. This research examines reports involving horse-drawn vehicles found in the Ag-InjuryNews dataset, which provides a publicly accessible collection of agricultural injury reports primarily gathered from news media. The goals of this research are to characterize the reports and to compare results with previous research to assess the utility of using AgInjuryNews to examine horse-drawn vehicle incidents. A total of 38 reports representing 83 victims were identified. Chi-square tests comparing victim and incident traits for fatal and nonfatal injuries were significant for the victim's role in the incident, vehicle type, presence of a motor vehicle, rear-ending by a motor vehicle, spooked horses, a victim being run over or struck by a vehicle, and a victim being ejected or falling from a vehicle. Additional analysis of incidents involving horse-drawn farm equipment showed that a significantly higher proportion of off-road incidents were fatal compared to on-road incidents. The proportion of fatal injuries in the AgInjuryNews dataset was approximately 10 times higher than observed in a study using Pennsylvania Department of Transportation (DOT) data. Compared to previous research, the AgInjuryNews reports contained a higher proportion of incidents where a motor vehicle rear-ended a horse-drawn vehicle, and fewer cases of horse-drawn vehicles being struck by motor vehicles while crossing or entering a main road and making left turns. Reports of buggy crashes found in AgInjuryNews differed from those found in a Nexis Uni search in that the bulk of the articles from Nexis Uni referred to cases involving criminal charges for impaired driving or hit-and-run crashes. While it is evident that the reports included in the sample are incidents that media sources find compelling rather than comprehensive injury surveillance, it is possible to gain new insights using the AgInjuryNews reports.

**Keywords:** roadway safety; horse-drawn; Anabaptist; buggy crash; AgInjuryNews

## 1. Introduction

Farming and ranching are some of the most dangerous occupations in the United States [1]. Still, agricultural injury surveillance has been an ongoing challenge, with a number of clear gaps and opportunities for improvement [2]. The surveillance of this industrial sector becomes even more complicated when trying to assess injuries and fatalities in unique populations within rural communities, including members of Old Order Anabaptist communities, such as the Amish, Old Order Mennonites, and Old Order Brethren. While the Centers for Disease Control and Prevention's National Occupational Research Agenda for Agriculture, Forestry, and Fishing does not specifically mention Anabaptist communities in their list of vulnerable populations, it does mention the need for research and targeted interventions for sub-sectors that may face unique challenges [3]. Anabaptist communities are not considered separately in the census of fatal occupational injuries (CFOI) [4], and cultural differences can complicate both surveillance programs and injury prevention strategies [5]. While the main focus of this research is injuries to members of Anabaptist groups in the United States, the insights gained may also benefit motor vehicle drivers in the US who regularly interact with horse-drawn vehicles, or drivers in other areas of the world where horse-drawn vehicles are common.

The term "Anabaptist" refers to members of a subset of Protestant religious groups that trace their origins back to the 16th century Protestant Reformation. Key beliefs held in common by these groups include baptism and church membership being limited to adult believers, nonresistance, and the separation of church and state [6]. While there are considerable variations among Anabaptist communities, a subset of these groups can be considered "Old Order," which means that they adhere to older forms of worship and limit their use of technology, including motorized vehicles [7]. The largest family of the Old Order Anabaptist groups is the Old Order Amish, who in 2021 were estimated to have over 355,000 members, predominantly in Pennsylvania, Ohio, Indiana, Wisconsin, and New York [8]. The Amish are one of the most rapidly growing subcultures in the US [9], nearly doubling from approximately 180,000 individuals in 2000 [10].

While fewer Amish today derive the bulk of their income from farming and the proportion of farmers varies considerably by community, it is estimated that around 10% are full-time farmers [11,12], compared with less than 2% of the general population [13]. Many of those who do not earn their primary income from farming and non-working family members still participate in agricultural activities, train and care for the horses they rely on for transportation, and live in rural areas [5]. Much of the population, therefore, faces the risks associated with agricultural production and horse-drawn vehicles, even if their primary income comes from another source.

The widespread use of animal- or horse-drawn vehicles and farm equipment results in a risk exposure profile that is very different from most US farmers and rural residents. Safety features that have reduced injuries and fatalities in motor vehicles and tractors, such as rollover protective structures, seatbelts, and, in the case of passenger vehicles, air bags, are rarely or never present on horse-drawn equipment or buggies [14]. Even when safety equipment, such as slow-moving vehicle signs, turn signals, and rear lights, can be installed, Anabaptist communities may resist using them because some groups forbid the use of electricity, while for others the bright colors and lights conflict with religious doctrine against drawing attention to oneself [14]. Relying on animals to pull buggies or farm equipment adds another layer of unpredictability inherent in animal behavior, especially when interacting with various motorized vehicles on public roadways.

On-road interactions between horse-drawn and motorized vehicles can pose additional hazards for members of Anabaptist communities and rural drivers alike. Buggy crashes, which often also involve motor vehicles, were the second most common source of traumatic injury to members of Anabaptist communities after falls, in Pennsylvania between 2000 and 2020 [15,16]. About 18.9% of nonfatal traumatic injuries reported involved buggy crashes [15]. A similar trauma center study conducted in Indiana between 1996 and 2002 identified 42 patients admitted after buggy crashes, 36 (86%) of whom were involved in crashes between a buggy and a motorized vehicle [17]. This study also examined crashes by month and found the highest number of crashes in July, October, and January [17]. In another study that monitored Pennsylvania Department of Transportation (DOT) crash reports from 2011 through 2013, 344 crashes were reported involving farm equipment and 246 involving buggies [18]. It was estimated that there were 198.4 farm equipment crashes and 89.4 buggy crashes per 100,000 Amish residents per year in Pennsylvania. The fatality rates for these crashes were 4.1% for farm equipment and 2.6% for buggies, which is much higher than the 1% fatality rate for all crashes involving any type of vehicle in Pennsylvania during that time. Rear-end and angle collisions were the most common [18]. This study also reported that 43.1% of motor vehicle drivers and 37.1% of motor vehicle passengers involved in a collision with Anabaptist farm equipment experienced injuries. For buggy crashes, 18.6% of motor vehicle drivers and 24.6% of motor vehicle passengers were injured [18].

A series of three studies published in the *Journal of Amish and Plain Anabaptist Studies* examined different facets of buggy crashes [14,19,20]. The first study ("Horse and Buggy Crash Study I") examined crash scenarios between motor vehicles and buggies using Pennsylvania DOT crash reports in 2006. Of the 77 cases examined, 31 involved motorists

rear-ending buggies, eight involved passing, 12 involved intersections or entering the road, and nine involved left turns [14]. The second study is a qualitative discussion of the usefulness of slow-moving vehicle signs and cultural barriers that can discourage their use [19], and the third examined the volume of buggy traffic and number of crashes by time, resulting in a profile of crash risk at different times of day [20]. It is worth noting that these three papers, as well as those mentioned earlier rely on case counts, categorization, and qualitative assessment rather than formal statistical analysis [14–20].

At least one previous attempt has been made at using newspaper articles to address injuries in Amish and Old Order Mennonite communities. Purdue University, in conjunction with the Centers for Disease Control and Prevention, the National Institute for Occupational Safety and Health, the Young Center for Anabaptist and Pietist Studies, and Elizabethtown College, created the Old Order Anabaptist Injury Database [21]. This database collects articles about Anabaptist injuries and fatalities through three Anabaptist publications: *Die Botschaft*, *The Budget*, and *The Diary* [21]. This data collection methodology identified 1153 cases in 2002, compared to an average of 28 cases per year identified by Google Alerts for 2011 and 2012 [22]. This dataset is not publicly available, nor is it possible to read the original publications without a subscription to obtain a printed copy. *The Budget*, which also includes both content written by less conservative Anabaptist groups and non-Anabaptist local news [22,23], has a Facebook page, but does not post full articles online [23].

One publicly accessible source of information that includes injuries and fatalities in Old Order Anabaptist communities that has not yet been examined in detail is the AgInjuryNews collection. AgInjuryNews, launched in 2015, is an online collection of news articles and other reports of agricultural injuries and fatalities. AgInjuryNews is curated by a team co-funded through the National Farm Medicine Center (NFMC) and the National Children's Center for Rural and Agricultural Health and Safety (NCCRAHS) [24]. As of 20 July 2022, the database contains 4603 reports involving 5141 victims. AgInjuryNews identifies articles primarily through a media subscription service, Google alerts, and submissions from colleagues. Articles meeting inclusion criteria are coded and entered into the system [25]. Each report contains pre-coded information about each incident such as the date, victim demographics, and a brief event description. Links to the original article or record are also contained in the incident report [26]. Additional information is typically available by request [26]. The AgInjuryNews reports have generated over a dozen publications, some focused on methodology [25,27,28], others on state-level surveillance [29–31], and others on specific topics of interest [32–34]. AgInjuryNews presents a unique opportunity to researchers in that the dataset is publicly accessible for free, and much of the work gathering and categorizing news media reports has been completed. While the dataset is subject to the inherent limitations of relying on media reports for case data and of the ability of the subscription service in identifying relevant articles, the ease of accessing and identifying media reports on a range of topics related to agricultural injuries is incentive to examine how this data source compares with traditional agricultural injury prevention research and where information from the dataset might be leveraged to reduce agricultural injuries and fatalities.

This paper has two primary goals. The first is to expand on the characterization of the AgInjuryNews dataset by examining reports involving animal- and horse-drawn Anabaptist vehicles. This will also allow for an assessment of the utility of AgInjuryNews for surveillance of Old Order Anabaptist injuries and fatalities and establish a baseline to which future data from AgInjuryNews can be compared. The second goal is to make a comparison between these reports and those recorded in previous studies to see how similar the AgInjuryNews results are to those of studies that used more traditional methodologies to investigate Anabaptist vehicle crashes.

## 2. Methods

### 2.1. Case Identification and Data Coding

Incident reports were sourced from AgInjuryNews. Reports were identified by using the "horse-drawn" filter. A total of 40 incidents involving 85 victims were initially identified. Of these, two single-victim incidents were excluded from the dataset because they did not involve members of Anabaptist communities, leading to a total of 38 incidents and 83 victims used in the analysis. The remaining incidents took place between 28 January 2016 and 22 March 2022. One incident involved a mule (half horse half donkey) and the other 37 incidents involved horses. Linked articles were available for 25 (66%) of the reports. Of the 13 (34%) of reports where the article was not accessible, 9 (24%) contained expired links, 1 (2%) contained a link that was active but behind a paywall, and 3 (8%) did not contain a link.

Each report and associated article were examined and relevant parameters for each incident were coded by the author. Victim demographics included age, gender, role in incident, and whether the injury was fatal or nonfatal. Incident parameters included date, state where the incident occurred, time of day, and number of victims. Nine potential contributing factors to the incident and injury severity were examined. These were whether the victim was ejected or fell, whether the victim was run over or struck by the vehicle, presence or absence of a motor vehicle, drug or alcohol use, spooked horses, rear-ending by another vehicle, passing by another vehicle, a left turn, and youth vehicle operators. Due to the way incidents were reported, it was often not possible to distinguish between falls and ejections or a victim being struck or run over by a vehicle. Therefore, these categories were grouped for analysis. The presence of slow-moving vehicle signs was coded but not included in the analysis because only one of the incident reports gave a clear indication of whether a slow-moving vehicle sign was used. Intersections were dropped from the first phase of analysis because no incidents took place at an intersection.

### 2.2. Victim and Incident Characteriesics

The first phase of analysis was to characterize the reports. For most variables relating to victim demographics and incident characteristics, a chi-squared analysis comparing counts of fatal and nonfatal injuries was performed. These variables were assessed at the victim level, i.e., each victim was a single data point. Data for time of day, month, and number of victims were summarized because either chi squared analysis was not appropriate or because other ways of presenting these data made more sense. For these variables, incident-level data were used, i.e., each data point represented a single incident.

### 2.3. Comparison with Other Data

In the second phase of analysis, data from AgInjuryNews were compared with other datasets. The first comparison was made between the number of reports and victims reported for each state in the AgInjuryNews dataset and the Amish population of those states.

The AgInjuryNews dataset was then re-coded for comparison with the Horse and Buggy Crash Study I dataset. The Horse and Buggy Crash Study I [14] was chosen for comparison because the classification of crashes into broad categories such as incidents involving a buggy being rear-ended by a motor vehicle and incidents taking place at intersections were found to be compatible with the information available in the AgInjuryNews reports. Since this study only included on-road crashes, off-road incidents were excluded from the AgInjuryNews dataset. Incident-level data were used for this assessment and each incident was coded to match the categories given in Table 1 of the Horse and Buggy Crash Study I results. Comparisons of the dates and times incidents occurred were made with the Pennsylvania DOT study [17] and Horse and Buggy Crash Study III [20] because these studies examined those parameters.

**Table 1.** Characteristics of victims and vehicles in AgInjuryNews dataset.

| Category | Fatal (*n* = 24) | Nonfatal (*n* = 59) | Total * (*n* = 83) | Chi-Square ** |
|---|---|---|---|---|
| Age | | | | |
| Child (age < 18) | 12 (50%) | 30 (51%) | 42 (51%) | X^2 = 0.2 |
| Adult (age ≥ 18) | 12 (50%) | 24 (41%) | 36 (43%) | dof = 1 |
| Unknown (excluded from x^2) | 0 (0%) | 5 (8%) | 5 (6%) | p = 0.6 |
| Gender | | | | |
| Female | 6 (25%) | 8 (14%) | 14 (17%) | X^2 = 0.01 |
| Male | 17 (71%) | 21 (36%) | 38 (46%) | dof = 1 |
| Unknown (excluded from X^2) | 1 (4%) | 30 (51%) | 31 (37%) | p = 0.9 |
| Role | | | | |
| Operator Buggy | 4 (17%) | 11 (19%) | 15 (18%) | |
| Passenger Buggy | 9 (38%) | 32 (54%) | 41 (49%) | X^2 = 9.6 |
| Operator Motorized Vehicle | 1 (4%) | 3 (5%) | 4 (5%) | dof = 4 |
| Operator Farm Equipment | 10 (42%) | 6 (10%) | 16 (19%) | p = 0.05 |
| Child Bystander | 0 (0%) | 1 (2%) | 1 (1%) | |
| Unknown (excluded from X^2) | 0 (0%) | 6 (10%) | 6 (7%) | |
| Vehicle Type | | | | |
| Buggy | 13 (54%) | 49 (83%) | 62 (75%) | X^2 = 7.5 |
| Horse-Drawn Farm Equipment | 11 (46%) | 10 (17%) | 21 (25%) | dof = 1 p = 0.006 |

* Column percentages may not add to 100 due to rounding errors. ** Unknown variables were not included in chi-square analysis.

An additional comparison was made between buggy crashes reported in AgInjuryNews and in news articles identified through Nexis Uni with the goal of comparing AgInjuryNews reports to a "standard" source used in news media research. Nexus Uni is a digital research database produced by Lexis Nexis and tailored for the needs of university faculty and student researchers. The database contains printed news articles, journals, television and radio broadcasts, newswires, blogs, legal documents, business information, and other document types from over 17,000 sources [35]. This source was selected as a second media source because Lexis Nexis has an extensive history of use for legal [36,37] and news media research [38,39]. Nexis Uni was the version of Lexis Nexis available at the author's institution. In order to capture as many articles related to buggy crashes as possible, a search of the term "buggy" with a secondary term of "crash" or "accident" were used to identify articles published between January 2016 and March 2022 in order to correspond to the date range. This search yielded a total of 105 results. Of these, 27 were excluded, due to reporting legal action for cases outside of the desired time interval (8 items), having a topic other than reporting a specific buggy crash (7 items), being a headline list rather than a complete article (8 items) or reporting incidents that took place outside of the U.S. (4 items). Many of the remaining articles referred multiple times to the same incident, with as many as 12 references to the same event. When these were consolidated, a total of 29 buggy crashes representing 96 victims were identified.

## 3. Results

### 3.1. Victim and Incident Characteriesics

A total of 38 incidents representing 24 fatal and 59 nonfatal injuries were identified in the AgInjuryNews dataset. Persons experiencing any nonfatal injury, no matter how minor, were included in the "nonfatal injury" category. Persons who were involved in incidents and were not injured were not analyzed because it was often unclear in both the articles and the reports whether additional uninjured victims were present. Fifteen incidents making up 39% of the total, involved multiple victims. As many as nine people were injured in a single event. Many of the victims in the larger incidents were children, and involved crashes between motor vehicles and buggies transporting entire Anabaptist families. All but two

of the 18 incidents involving farm equipment and wagons were single-victim incidents, compared to just seven out of the 20 buggy incidents.

An average of 3.2 incidents took place each month. June and December had the most incidents with five each, and January and May had the fewest with one each. These results are somewhat different than those reported in the Pennsylvania DOT study [3], which found the most incidents in July, October, and January, and no incidents in April and June. As for the time of day, incidents were split somewhat evenly. A total of 12 incidents took place in the morning (6:00 am–11:59 am), eight in the afternoon (12:00 pm–5:59 pm), 11 in the evening (6:00 pm–11:59 pm), and seven at a time that was not specified. Horse and Buggy Crash Study III noted peaks in crash incidents around 8:00 am and 6:30 pm [4].

Victim age, victim gender, victim role, type of vehicle involved, and chi-squared results can be seen in Table 1 below. Chi-squared results were significant regarding victim role and vehicle type. A higher percentage of horse-drawn farm equipment operators experienced fatal injuries than those injured in buggy crashes, while injuries to motor vehicle operators involved in these crashes were rare. The single motor vehicle operator fatality was a motorcyclist. The percentage of fatal injuries was 23% for members of the Anabaptist community injured in buggy crashes and 59% for horse-drawn farm equipment incidents, much higher than was the rate of 2.6% and 4.1%, reported in the Pennsylvania DOT study [5].

In light of the increased proportion of fatalities observed in the 21 horse-drawn equipment incidents, further analysis was performed to compare on-road and off-road incidents. Of the eight (38%) victims injured in on-road incidents, two (25%) were fatally injured and six (75%)were nonfatally injured. In contrast, of the 13 (62%) off-road incidents, nine (69%) were fatally injured and four (31%) were nonfatally injured. A chi-square test found that the difference was significant ($X^2 = 3.9$ dof = 1 $p = 0.05$).

Incident characteristics, including the involvement of motor vehicles, alcohol or illegal drugs, rear-ending by a motor vehicle, spooked horses, a fall or ejection from vehicle, being run over or struck by vehicle, passing by a motor vehicle, a left turn, distracted driving, and youth driver or operator, can be seen in Table 2 Results were significant for involvement of motor vehicles, rear-ending by a motor vehicle, spooked horses, being run over or struck by a vehicle, and fall or ejection from vehicle. A fall or ejection may or may not have been paired with being run over or struck by a vehicle, so those factors were treated separately. Some example scenarios found in the dataset for these variables include: a child running in front of a piece of farm equipment and being run over, a buggy driver being thrown clear of a crash with a motor vehicle without being struck or run over, a buggy passenger ejected and then run over by a motor vehicle. In some cases, the level of detail of the article and/or the known information about the case made it unclear whether the victim was struck vs. run over, or fell vs. was ejected, which is why these variables were paired. Incidents involving a motor vehicle, including those where the motor vehicle rear-ended the Anabaptist vehicle, were less likely to be fatal. Spooked horses, being run over or struck by a vehicle, falling, or being ejected from a vehicle were more likely to be fatal.

### 3.2. Comparison with Other Data

A summary of the number of incident reports and victims for each state, the number of incidents and victims per 100,000 Amish residents, the Amish population of each state as sourced from the Young Center for Anabaptist and Pietist Studies [8], and the rank of each state in terms of Amish population, with 1 being the state with the largest Amish population, can be seen in Table 3. It is important to remember that these are the reporting rates for the AgInjuryNews database specifically, and should not be interpreted as real-world incidence rates. While other Old Order Anabaptist communities besides the Amish use horse-drawn vehicles and appear in the AgInjuryNews dataset, separate population counts for these groups were not available and the Amish are by far the largest group both in the dataset and in terms of population. The states with the greatest number of incident reports generally corresponded well to the states with the highest Amish population, with

four of the top five in terms of number of reports also being in the top five in population. However, when examined in terms of articles per 100,000 Amish residents, states with low Amish populations, such as Iowa and Minnesota, are over-represented.

**Table 2.** Incident characteristics from AgInjuryNews dataset.

| Category | Fatal (*n* = 24) | Nonfatal (*n* = 59) | Total * (*n* = 83) | Chi-Square ** |
|---|---|---|---|---|
| Motor Vehicle Involved | | | | |
| Yes | 15 (62%) | 55 (93%) | 70 (84%) | X^2 = 12.2 dof = 1 |
| No | 9 (38%) | 4 (7%) | 13 (16%) | *p* = 0.0005 |
| Alcohol or Illegal Drugs Involved | | | | |
| Yes | 1 (4%) | 9 (15%) | 10 (12%) | X^2 = 2.0 |
| No | 23 (96%) | 50 (85%) | 73 (88%) | dof = 1 |
| | | | | *p* = 0.2 |
| Rear-Ended by Motor Vehicle | | | | |
| Yes | 10 (42%) | 36 (61%) | 46 (55%) | X^2 = 9.1 |
| No | 11 (46%) | 7 (12%) | 18 (22%) | dof = 1 |
| Unknown | 3 (12%) | 16 (27%) | 19 (23%) | *p* = 0.003 |
| Spooked Horse | | | | |
| Yes | 7 (29%) | 5 (8%) | 12 (14%) | X^2 = 10.0 |
| No | 14 (58%) | 34 (58%) | 48 (58%) | dof = 1 |
| Unknown | 3 (12%) | 20 (34%) | 23 (28%) | *p* = 0.002 |
| Run Over or Struck by Vehicle | | | | |
| Yes | 7 (29%) | 5 (8%) | 12 (14%) | X^2 = 3.5 |
| No | 14 (58%) | 34 (58%) | 48 (58%) | dof = 1 |
| Unknown | 3 (12%) | 20 (34%) | 23 (28%) | *p* = 0.06 |
| Fell or Ejected from Vehicle | | | | |
| Yes | 12 (50%) | 9 (15%) | 21 (25%) | X^2 = 4.4 |
| No | 0 (0%) | 4 (7%) | 4 (5%) | dof = 1 |
| Unknown | 12 (50%) | 46 (78%) | 28 (70%) | *p* = 0.04 |
| Passing by Motor Vehicle | | | | |
| Yes | 0 (0%) | 1 (2%) | 1 (1%) | X^2 = 0.5 |
| No | 21 (88%) | 42 (71%) | 63 (76%) | dof = 1 |
| Unknown | 3 (12%) | 16 (27%) | 19 (23%) | *p* = 0.5 |
| Horse-Drawn Vehicle Left Turn | | | | |
| Yes | 1 (4%) | 1 (2%) | 2 (2%) | X^2 = 0.3 |
| No | 20 (83%) | 42 (71%) | 62 (75%) | dof = 1 |
| Unknown | 3 (12%) | 16 (27%) | 19 (23%) | *p* = 0.6 |
| Distracted Motor Vehicle Driver | | | | |
| Yes | 1 (4%) | 8 (14%) | 9 (11%) | X^2 = 1.8 |
| No | 21 (88%) | 42 (71%) | 63 (76%) | dof = 1 |
| Unknown | 2 (8%) | 9 (15%) | 11 (13%) | *p* = 0.2 |
| Youth (>18) Operator of Any Vehicle | | | | |
| Yes | 6 (25%) | 8 (14%) | 14 (17%) | X^2 = 1.3 |
| No | 18 (75%) | 48(81%) | 66 (80%) | dof = 1 |
| Unknown | 0 (0%) | 3 (5%) | 3 (4%) | *p* = 0.2 |

* Column percentages may not add up to 100 due to rounding errors. ** Unknown variables were not included in chi-square analysis.

A comparison of on-road crashes between the AgInjuryNews dataset and those reported in Horse and Buggy Crash Study I can be found in Table 4. The AgInjuryNews dataset contained more reports involving a motorist rear-ending an Anabaptist vehicle, and no reports of incidents that occurred when an Anabaptist driver tried to cross or enter a main road. The chi-square results were not quite significant at the *p* < 0.05 level.

**Table 3.** Number of AgInjuryNews incidents and injuries per state and 2021 in Amish population.

| State | Number of Incidents in AgInjuryNews Dataset | Number of Injuries in AgInjuryNews Fatal (Nonfatal) | Incidents Recorded in AgInjuryNews per 100,000 Amish Residents | Injuries Recorded in AgInjuryNews per 100,000 Amish Residents Fatal (Nonfatal) | Amish Population From Young Center Data | Amish Population Rank |
|---|---|---|---|---|---|---|
| Wisconsin | 11 | 6 (26) | 47.4 | 25.9 (112.1) | 23,195 | 4 |
| New York | 6 | 4 (5) | 27.6 | 18.4 (23.0) | 21,725 | 5 |
| Ohio | 5 | 3 (9) | 6.23 | 3.7 (11.2) | 80,240 | 2 |
| Missouri | 4 | 2 (6) | 27.4 | 13.7 (41.1) | 14,610 | 7 |
| Pennsylvania | 4 | 5 (1) | 4.8 | 5.9 (1.2) | 84,100 | 1 |
| Iowa | 3 | 0 (3) | 30.5 | 0 (30.5) | 9845 | 9 |
| Michigan | 2 | 3 (8) | 11.3 | 17.0 (45.2) | 17,695 | 6 |
| Indiana | 2 | 0 (1) | 3.28 | 0 (1.6) | 60,960 | 3 |
| Minnesota | 1 | 1 (0) | 20.3 | 20.3 (0) | 4935 | 10 |

**Table 4.** Comparison of Horse and Buggy Crash Study I results and AgInjuryNews data.

| Category | Horse and Buggy Crash Study I | AgInjuryNews | Chi-Squared |
|---|---|---|---|
| Motorist rear-ended a forward-moving Anabaptist vehicle | 31 (41%) | 16 (57%) | |
| Motorist attempted to pass a forward-moving Anabaptist vehicle | 8 (10%) | 1 (4%) | |
| Anabaptist driver attempted to cross or enter a main road | 12 (16%) | 0 (0%) | $X^2 = 7.4$ $dof = 3$ $p = 0.06$ |
| Anabaptist driver attempted a left turn off the main road | 9 (12%) | 2 (7%) | |
| Other small categories or unknown * | 16 (21%) | 9 (32%) | |
| Total | 70 | 28 | |

* The "other or unknown" category was not included in chi-square analysis.

A final comparison was made between the 20 buggy crashes found in the AgInjuryNews dataset and 29 buggy crashes identified in Nexis Uni. A total of five incidents (25% of the AgInjuryNews reports, 17% of the Nexis Uni reports) appeared in both datasets. While the goal of this comparison was to use the Nexis Uni reports as a standard to which the AgInjuryNews reports could be compared, it quickly became evident during analysis that the two data sources had different intents and a characterization of the differences between the two sources was more appropriate. It was observed that many of the reports gathered from Nexis Uni documented ongoing legal proceedings for criminal cases involving impaired driving or hit-and-run in association with buggy crashes, making up a total of 43 of the 78 relevant articles identified in the Nexus Uni dataset, representing 11 incidents. Several additional articles mentioned investigations that did not ultimately lead to charges. These cases were largely absent from the AgInjuryNews reports. A comparison of the datasets with the five cases that appeared in both omitted can be seen in Table 5. The results of the chi-squared test were significant only when comparing the total number of articles related to incidents involving criminal cases. The difference in proportion of incidents involving the two types of criminal charges noted in the Nexus Uni dataset is also documented.

**Table 5.** Comparison of incident characteristics between AgInjuryNews and Nexis Uni.

|  | AgInjuryNews (*n* = 15) | Nexis Uni (*n* = 24) | Total (*n* = 39) | Chi-Square |
|---|---|---|---|---|
| Alcohol or Drug Impairment |  |  |  |  |
| yes | 1 (7%) | 6 (25%) | 7 (18%) | X^2 = 2.1 |
| no | 14 (93%) | 18 (75%) | 32 (82%) | dof = 1 |
|  |  |  |  | *p* = 0.15 |
| Hit and Run |  |  |  |  |
| yes | 0 (0%) | 5 (21%) | 5 (13%) | X^2 = 3.5 |
| no | 15 (100%) | 19 (79%) | 34 (87%) | dof = 1 |
|  |  |  |  | *p* = 0.06 |
| Any Criminal activity |  |  |  |  |
| yes | 1 (7%) | 11 (46%) | 12 (31%) | X^2 = 6.6 |
| no | 14 (93%) | 13 (54%) | 27 (69%) | dof = 1 |
|  |  |  |  | *p* = 0.01 |

## 4. Discussion

Incident reports involving horse-drawn Anabaptist vehicles from the AgInjuryNews dataset were successfully coded and characterized. The results of this characterization revealed some of the strengths and limitations of this data source. One of the key advantages of this data source is public accessibility. While Purdue's newspaper surveillance system that targets Anabaptist publications yields over one thousand records annually, neither those records nor the articles that generated them are accessible to outside researchers. Moreover, while the AgInjuryNews dataset is publicly accessible, fewer than 10 incidents involving horse-drawn Anabaptist vehicles per year have been found therein. This is perhaps an indication that there would be a benefit to expanding the ways in which articles are gathered by the AgInjuryNews team, or an indication of an inherent limitation of the system, if these articles are not included due to copyright limitations.

The proportion of fatal cases in the AgInjuryNews data was much higher than that reported in the Pennsylvania DOT study. This perhaps shows the limitations of using news media reports for injury surveillance, as the more severe are more likely to be covered by news media outlets, particularly incidents occurring on public roadways, and those involving multiple victims. Knowing that the percentage of fatal injuries in the AgInjuryNews dataset is over ten times higher than those reported in a study that performed a comprehensive evaluation of all DOT reports involving Old Order Anabaptist vehicles in the state with the highest Amish population provides one clear estimate of the extent to which severe incidents are overrepresented. This tendency for overrepresentation of severe incidents is a limitation mentioned by previous researchers as well, who have added that these types of traumatic (the most serious) injury events are the ones we want most to prevent [40]. Additionally, the inclusion of victims who were not injured in the articles and in the AgInjuryNews reports was sporadic at best. There was at least one case where two additional survivors of a fatal incident were not mentioned in the primary article, but were mentioned in a linked follow-up article. In other cases, victims were labeled as "uninjured" despite also being described as suffering cuts and bruises. At times it was also difficult to distinguish between whether an uninjured person should be described as a victim of the incident, or as a witness, bystander, or help. For instance, if an article mentions that a family member stopped a spooked horse without injury after the driver had fallen from a piece of horse-drawn machinery, should they be considered an uninjured victim of the incident? More research is needed to see if these trends hold for other parts of the AgInjuryNews dataset.

The overrepresentation of severe incidents is particularly evident in the off-road incidents involving horse-drawn farm equipment. While the proportion of fatal on-road incidents was comparable to the proportion of fatalities observed in buggy crashes, nearly 70% of off-road equipment incidents were fatal. It is possible that on-road incidents are more likely to attract the attention of emergency services and non-Anabaptist residents, increasing the likelihood that the story would be covered by mainstream media sources.

Additionally, all multi-victim incidents observed in the AgInjuryNews dataset occurred on-road. The presence of multiple victims may simultaneously increase the likelihood of an incident being reported in media and decrease the rate of fatalities reported in on-road incidents since a combination of fatal and nonfatal injuries is common. In any case, this further enforces the role media selection of what incidents to cover plays in the AgInjuryNews dataset.

The analysis of the AgInjuryNews dataset also reveals a set of factors related to increased fatalities that is somewhat different from those mentioned in other studies. Factors such as spooked horses, being ejected from the vehicle, or being run over and struck by a vehicle seem to have a stronger association with fatalities than the road layout and traffic patterns that have been the focus of other research. Given that the selection of cases that appear in the dataset is determined by media sources and not a systematic surveillance process, it is hard to say for sure to what extent these observations represent real-world trends; however, perhaps both researchers and makers of Anabaptist vehicles need to look more at crash protection features in the vehicle itself rather than the types of interactions happening on the road.

Another limitation of the system is that while states with a higher Amish population generally produced more reports, this trend disappears when examined as a rate of number of reports per 100,000 Amish residents. In some states, where the Amish population is very low, such as Iowa or Indiana, the presence of only one or two reports results in a very high rate. This could be simply an effect of the small sample size, or it could be an indication that incidents involving Anabaptist vehicles are more likely to receive media attention in states with fewer Anabaptist residents. States with higher numbers of reports in the AgInjuryNews database generally corresponded to states with more injury reports in the AgInjuryNews database. For all states except Minnesota and Pennsylvania, which only had one and six reported injuries, respectively, the rate of reported nonfatal injuries was greater than the rate of reported fatal injuries. This shows that even though fatal and severe incidents are overrepresented in media reports, the expectation that the rate of nonfatal injuries is higher than that of fatal injuries holds. It is important to note that since AgInjuryNews misses many media articles involving Anabaptist vehicle incidents and likely only captures only a small percentage of all incidents that occur, it is not reasonable to believe that the rates reported in the database represent real-world incidence rates. The real-world rates would presumably be higher. The database rates are useful in their own right, however, because they give a population-adjusted measure of reporting by state and can serve as a more reliable baseline for comparison with future AgInjuryNews data since this is a rapidly growing population.

Examining the month and time of day when the incidents took place provides another opportunity for comparison with previous research. The distribution of incidents was fairly even, with 2–4 incidents reported during most months. This contrasts with the results of the Pennsylvania DOT study, which noted dramatic differences in the number of incidents from month to month. Again, this is perhaps an indication that the timing of media reports might differ from real-world incidence rates. The interpretation of the time of day when incidents took place is somewhat hampered by the fact that AgInjuryNews only reports the time incidents took place in 6-h blocks, and the original articles rarely allow for a more precise determination of time. Horse and Buggy Crash Study III [20] hypothesizes that the risk for crashes is greatest at the transitions between daylight and darkness, which might also correspond to times when motorist traffic is greatest, as people commute to and from work. One of these transitional periods takes place between 5:00 pm and 7:00 pm, meaning that corresponding data would be split between the "Afternoon" and "Evening" categories used by AgInjuryNews. The nature of the two datasets makes direct comparisons difficult, but this may be one area where the AgInjuryNews dataset more closely follows other data sources, since splitting the afternoon transitional period would partially cancel out the lower number of incidents taking place in the early afternoon.

While there is not a good comparison point in the literature for the number of victims per incident, these results highlight how severe these incidents can be, and is also a reminder of how frequently children are being transported in these vehicles on public roadways. In one of the most memorable cases of the dataset, eight children were injured and their mother killed after their buggy was rear-ended by a motorist under the influence of illegal drugs. This is perhaps another piece of evidence in favor of shifting the focus towards re-designing Anabaptist vehicles in order to prevent injuries during a crash. These results also help to illustrate the differences between buggy crashes and incidents involving horse-drawn farm equipment.

The comparison between the AgInjuryNews dataset and the results of the Horse and Buggy Crash Study I shows further differences between AgInjuryNews and studies with more comprehensive injury surveillance. While these results were not quite significant, the AgInjuryNews dataset contains more incidents involving an Anabaptist vehicle being rear-ended by a motor vehicle and fewer incidents involving Anabaptist vehicles being struck while crossing or entering a road and while making left turns.

The comparison between media reports gathered through AgInjuryNews and media reports identified in Nexis Uni provides another important facet for determining how the AgInjuryNews dataset compares to traditional research methodologies. The intent of this comparison was to set AgInjuryNews up against a "standard" for news-media-based research using a database that was presumed to have a more complete record. Rather than being able to use Nexis Uni to establish the level of "completeness" of the AgInjuryNews database, the findings instead indicate that the two systems collect very different types of data, and have relatively little overlap between the databases. This also indicates that each source is missing many relevant articles on this topic. Since most of the articles referenced in Nexis Uni are also a matter of public record, this is perhaps another area where AgInjuryNews could improve their data collection procedures to capture more cases.

It is also worth mentioning that the relatively small number of cases is a significant limitation in this study and in most Anabaptist vehicle research. This dataset was chosen for examination in part because it involved a special population that is often overlooked in roadway and in agricultural research. It was also chosen because the relatively small number of cases and narrow scope facilitated the analysis of the strengths and weaknesses of the AgInjuryNews database. Additional reports are added to the database regularly, so it will be worthwhile to return to this topic after some time has passed to see if the trends noted in this preliminary analysis hold true over time.

## 5. Conclusions

The analysis of horse-drawn Anabaptist vehicle incidents found in the AgInjuryNews dataset offers valuable insights into the strengths and limitations of the dataset, and the characteristics of these incidents. While it has long been known that this dataset is subject to the limitations of what media sources choose to publish rather than a comprehensive surveillance system, this research was able to quantify that the percentage of fatal injuries reported in the AgInjuryNews dataset was roughly ten times higher than that reported in more comprehensive surveillance studies. For horse-drawn farm equipment incidents that did not take place on public roadways, the overrepresentation of fatal incidents was even stronger. The lack of seasonal cycles, which have been observed in other studies in the AgInjuryNews reports, may hint at incidents being reported based on news cycles rather than real-world fluctuations in incidence rates, or may simply be an indication that horse-drawn vehicles are consistently utilized for transportation year round. More research is needed to see whether these trends remain constant across the AgInjuryNews dataset; however, it is an important step forward in being able to account for how the reports gathered by AgInjuryNews might differ from real-world incidents.

With regard to incident characteristics, the increased proportion of fatalities in victims who were struck by, run over by, fell from, or were ejected from vehicles may indicate that focusing on keeping drivers and passengers in their vehicles, rather than on the

characteristics of the crash, might make a greater difference in preventing fatalities. Still, only one of the reports included any mention of slow-moving vehicle (SMV) emblems, marking, or lighting, which should be clearly and consistently reported as a key injury prevention strategy. Interestingly, this analysis also uncovered that just short of half of the victims were buggy passengers. Furthermore, roughly half of the victims were children, and seventeen percent of incidents involved a teen driver of an Anabaptist or motorized vehicle. Some of these incidents involved up to nine members of the same family being injured or killed in a single incident, which can have lifelong physical and cognitive impacts on the victims, and mental health implications for everyone involved in these incidents and for the community. Further research is needed for the prevention of childhood injuries associated with horse-drawn equipment and buggies and to document the far-reaching consequences these incidents have on survivors and rural communities.

**Funding:** This research received no external funding. A summary of funding for AgInjuryNews, whose data were used in this research, can be found at About Us—AgInjuryNews—AgInjuryNews.

**Institutional Review Board Statement:** As this research used publicly available data primarily sourced through news media articles, no institutional review board oversight was sought for this research.

**Informed Consent Statement:** Informed consent was not applicable for this study because all data used came from news media articles and other forms of public record. Therefore, it is not considered to be human subjects research.

**Data Availability Statement:** All data sourced through AgInjuryNews are publicly available for free via their website, https://www.aginjurynews.org/ (accessed on 19 July 2022).

**Acknowledgments:** The author wishes to acknowledge the following individuals and institutions for their contributions to this research: Bryan Weichelt, Associate Research Scientist Marshfield Clinic Research Institute, for general reviewing/proofreading and checking that AgInjuryNews procedures were described correctly. Serap Gorucu, Assistant Professor University of Florida, for general reviewing/proofreading and for checking statistics and tables. Jeremy Swist, Lecturer Brandeis University, for initial proofreading and grammar-checking. The National Farm Medicine Center/The National Children's Center for Rural and Agricultural Health/Marshfield Clinic Research Institute/The AgInjuryNews Development and Support Team, these are the parent institutions of AgInjuryNews. Without their efforts to develop and maintain a publicly accessible online database of agricultural injury news articles, this research would not have been possible. The University of Southern Indiana, this is the author's home institution, without whose support, this research would not have been possible.

**Conflicts of Interest:** The authors declare no conflict of interest.

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
