# Peer review of "An Assessment of Horse-Drawn Vehicle Incidents from U.S. News Media Reports within AgInjuryNews"

_safety_

Round 1
Reviewer 1 Report
The paper presents a simple statistical analysis of accidents involving horse-drawn vehicles. The analyzed data comes from the AgInjuryNews dataset. AgInjuryNews is an online collection of news articles and other reports of agricultural injuries and fatalities.
I believe that the topic of the study is of poor importance to the scientific community dealing with road safety. Furthermore, the data on accidents are presented through a simple statistical survey, without a real scientific methodology.
Therefore I believe that the paper is not suitable for publication.
Author Response
Dear Reviewer 1
I appreciate the time and effort you have spent reviewing this article, An Assessment of Horse-Drawn Vehicle Incidents from U.S. News Media Reports within AgInjuryNews. Your feedback has led to substantial improvements in the article. I have carefully considered each item, and present my revisions in two formats. A brief response to your comment can be seen below. Additionally, changes made to the manuscript have been tracked and are highlighted in red. I hope the revisions have improved the quality of the manuscript, and welcome any further suggestions for improvement.
Item 1
Comment: I believe that the topic of the study is of poor importance to the scientific community dealing with road safety. Furthermore, the data on accidents are presented through a simple statistical survey, without a real scientific methodology.
Response: This paper deals with a special population. While the population of horse-drawn vehicle users in the US is relatively low, in many rural areas across the country motorized vehicle operators routinely encounter horse-drawn vehicles and are therefore also benefited by improvements in horse-drawn vehicle safety. A few additional sentences to this effect have been added to the introduction. There may also be international applications for this research in countries where interactions between horse-drawn and motorized vehicles are more common, as pointed out by one of the other reviewers. This research also serves as an example, utilizing a data subset with a smaller number of cases, of some of the key benefits and limitations of using AgInjuryNews as a data source.
Reviewer 2 Report
I offer a few possible suggestions that might be used to improve the likelihood of achieving the stated goals of this study with regard to Anabaptist article horse drawn carriages and equipment. This minority population is at increased risk of serious injury or death from road crashes and I applaud the efforts of this author to attempt to improve the data and information that might be used to inform preventative efforts. I offer some suggestions and approaches that might (or might not) be of interest in furthering the stated goals.
The stated goals of this study included:
1) expand on the characterization of the AgInjuryNews dataset by examining reports involving animal- and horse-drawn Anabaptist vehicles
2) assess the utility of AgInjuryNews for surveillance of Old Order Anabaptist injuries and fatalities,
3) establish a baseline to which future data can be compared.
4) make a comparison between these reports and those recorded in previous studies in order to see how similar the AgIn juryNews results and more traditional methodologies to investigate Anabaptist vehicle crashes.
In regard to the first goal, inaccessible data and news reports is a study limitation. There is a need for a stronger “gold standard” with which to compare the AgInjuryNews if it is being used as a baseline.
There is a free data set available for downloading on NHTSA website, FARS, that contains this information for the fatal crashes on U.S roadways that could be another valuable surveillance tool for investigating injury and fatalities. This would only capture on-road crashes, but use of Lexus Nexus newspaper reports could also be employed compared with the AgInjuryNews deaths on a roadway to see how complete the AgInjuryNews is. While it is not clear that one could identify anabaptist from the FARS data base, the agnews reported that approximately 95% of the horse and buggy/equipment crashes were found to be anabaptist so when combine with the GIS data, this might be a novel and informative approach.
The NHTSA data set has a lot more detail on the roadway where the crash occurred, GPS coordinates that allow mapping, the type MV vehicle that crashed, speed, impairment, age of the drivers, urban/rural environment etc. this data set is widely viewed as the gold standard for fatalities on U.S. roadways. It can be located here: https://www.nhtsa.gov/research-data/fatality-analysis-reporting-system-fars. This could be a stronger database for surveillance, comparison and establishing a baseline. It has GIS coding elements where the crashes could be mapped.
This article presents very interesting background on an important topic that is recognized as an important issue and is currently being addressed in these states and beyond (Iowa).
The article makes use of some interesting data, but could expand their analyses to make it more informative from both a surveillance and prevention standpoint. A weakness of the analysis is that not all news reports were accessed. Another possibility is the use of Lexus Nexus database of news reports. Since a stated goal is to conduct surveillance, have a baseline and examine injury and death, there might be stronger approaches for evaluating how complete the AgInjuryNews is and for establishing a baseline. It is also possible that the combination of multiple data sets would give a more complete picture.
Page 6, lines 226-230 are very misleading. It is stated that the states with the larger number of incidence reports were the states with the highest Anabaptist populations. Another way of viewing this data in Table 3 would be to pop the data into Excel and calculate the crash incident rates per 100,000 population—as I have done below. One can also see that the top 3 highest ranking states in terms of population have the lowest incidents of reported crashes per 100,000, suggesting possibly that there may be some ongoing prevention efforts (at least one state has both road signage and buggy signage prevention campaigns ongoing). (A possible typo--There is no rank of 10, but a rank of 11).
|
Rate per 100,000 |
Pop Rank |
State |
|
4.76 |
1 |
Pennsylvania |
|
6.23 |
2 |
Ohio |
|
3.28 |
3 |
Indiana |
|
47.42 |
4 |
Wisconsin |
|
27.62 |
5 |
New York |
|
11.30 |
6 |
Michigan |
|
27.38 |
7 |
Missouri |
|
30.47 |
9 |
Iowa |
|
20.26 |
11 |
Minnesota |
Author Response
Dear Reviewer 2
I appreciate the time and effort you have spent reviewing this article, An Assessment of Horse-Drawn Vehicle Incidents from U.S. News Media Reports within AgInjuryNews. Your feedback has led to substantial improvements in the article. The two additional data sources that you mentioned were particularly helpful, both for this research and as potential resources for future studies. I have carefully considered each item, and present my revisions in two formats. A brief response to each individual reviewer recommendation can be seen below. Additionally, changes made to the manuscript have been tracked and are highlighted in red. I hope the revisions have improved the quality of the manuscript, and welcome any further suggestions for improvement.
Item 1
Comment: There is a need for a stronger “gold standard” with which to compare the AgInjuryNews if it is being used as a baseline
Response: There is a need, and it is a challenging thing to accomplish, especially for this population. I was hoping that Nexus or FARS would work for this but neither source panned out in the way I would have liked. Hopefully having additional comparisons helps.
I also went back to clarify that by “baseline” I meant “baseline for comparison of future AgInjuryNews data,” and not in the general sense.
Item 2
Comment: There is a free data set available for downloading on NHTSA website, FARS, that contains this information for the fatal crashes on U.S roadways that could be another valuable surveillance tool for investigating injury and fatalities. This would only capture on-road crashes, but use of Lexus Nexus newspaper reports could also be employed compared with the AgInjuryNews deaths on a roadway to see how complete the AgInjuryNews is. While it is not clear that one could identify anabaptist from the FARS data base, the agnews reported that approximately 95% of the horse and buggy/equipment crashes were found to be anabaptist so when combine with the GIS data, this might be a novel and informative approach.
The NHTSA data set has a lot more detail on the roadway where the crash occurred, GPS coordinates that allow mapping, the type MV vehicle that crashed, speed, impairment, age of the drivers, urban/rural environment etc. this data set is widely viewed as the gold standard for fatalities on U.S. roadways. It can be located here: https://www.nhtsa.gov/research-data/fatality-analysis-reporting-system-fars. This could be a stronger database for surveillance, comparison and establishing a baseline. It has GIS coding elements where the crashes could be mapped
Response: I spent several days working with FARS and so far have not found a way to isolate the anabaptist cases or to match cases. Presumably anabaptist vehicles are coded under “farm equipment” and/or “other”. I also tried to find a way to match cases using other factors such as state, county, date, victim characteristics, etc. but the types of data provided by FARS and the types of data provided by AgInjuryNews weren’t giving me enough measures in common to make matches, at least so far as I was able to determine in the limited time allowed for revisions.
As a side note, I recently finished a draft of another paper for which I do have a means of case matching. That paper examines all agricultural fatalities in Indiana, with case matching performed between the AgInjuryNews cases and the Purdue extension’s annual reports of agricultural fatalities. What I’m finding is that there is a high, but not 100% overlap in fatal agricultural injuries between those data sources, and that for non-fatal injuries both sources are primarily capturing non-fatal injuries occurring alongside a fatal injury. I was not able to examine anabaptist cases separately due to how few cases there were in Indiana.
Item 3
Comment: The article makes use of some interesting data,but could expand their analyses to make it more informative from both a surveillance and prevention standpoint. A weakness of the analysis is that not all news reports were accessed
Response: The analysis was expanded using Nexus articles (see response to item 4)
As for the missing articles, this is an inherent limitation of the database.
Item 4
Comment: Another possibility is the use of Lexus Nexus database of news reports. Since a stated goal is to conduct surveillance, have a baseline and examine injury and death, there might be stronger approaches for evaluating how complete the AgInjuryNews is and for establishing a baseline. It is also possible that the combination of multiple data sets would give a more complete picture
Response: I was able to access Nexus Uni and perform additional analysis comparing their buggy-related incidents to those found in AgInjuryNews. Nexus Uni wasn’t the ”gold standard” we were hoping it would be because the types of cases that appear there are very different than the AgInjuryNews cases, but that knowledge in itself is valuable information.
Item 5
Comment: Page 6, lines 226-230 are very misleading. It is stated that the states with the larger number of incidence reports were the states with the highest Anabaptist populations. Another way of viewing this data in Table 3 would be to pop the data into Excel and calculate the crash incident rates per 100,000 population—as I have done below. One can also see that the top 3 highest ranking states in terms of population have the lowest incidents of reported crashes per 100,000, suggesting possibly that there may be some ongoing prevention efforts (at least one state has both road signage and buggy signage prevention campaigns ongoing). (A possible typo--There is no rank of 10, but a rank of 11). *note example chart was omitted for the sake of brevity*
Response: Chart and discussion updated according to recommendations
Reviewer 3 Report
The article touches on the issue of the accidents related to horse-drawn vehicles. An accident data analysis and discussion of results is reported.
I have some comments for the author.
I think that a missing analysis in the Introduction section concerns the review of the scientific literature about the previous studies carried out on the topic of “horse-drawn vehicle incidents”. There is a deep discussion of the term “Anabaptist” and of the several sources of accident data but the title of the study focuses on horse-drawn vehicle incidents: it could be useful for the reader a scientific background concerning the different approaches in literature (if any) for the analysis of these types of crashes. Bring the text in the introduction more in line/focused on the topic of the study.
On line 192 Reference number is missing.
In the guidelines for authors is reported: “Tables should be inserted into the main text close to their first citation”; I suggest to move Table 1 from section 3 to section where it is cited; moreover, there is no citation in the main text for Table 2 (I think that on line 211 Table 3 stands for Table 2).
Please improve the layout of Table 2: I suggest to divide each “category” in a separate row.
Line 220 and 235: Reference is missing.
Table 3 is not present in the main text.
Please provide insights (in the main text or as a note in the table) on the “Amish Population Rank” in Table 3 and how it is assessed
On line 226 you write: “The states with the greatest number of reports generally corresponded well to the states with the highest Amish population, with 4 of the top 5 in terms of number of reports also being in the top 5 in population.” Please review this quote. For example, Pennsylvania and Iowa have 4 and 3 reports, respectively, but the rank is 1 for Pennsylvania (84,100 population) and 9 for Iowa (9845 population). This means that a correlation between the number of reports and the population rank is not completely significant. I suggest reviewing the discussion of this data.
Best regards
Author Response
Dear Reviewer 3
I appreciate the time and effort you have spent reviewing this article, An Assessment of Horse-Drawn Vehicle Incidents from U.S. News Media Reports within AgInjuryNews. Your feedback has led to substantial improvements in the article. I have carefully considered each item, and present my revisions in two formats. A brief response to each individual reviewer recommendation can be seen below. Additionally, changes made to the manuscript have been tracked and are highlighted in red. I hope the revisions have improved the quality of the manuscript, and welcome any further suggestions for improvement.
Item 1
Comment: On line 192 Reference number is missing.
Response: Restored reference to Table 1
Item 2
Comment: Please improve the layout of Table 2: I suggest to divide each “category” in a separate row
Response: Re-formatted this table to improve layout
Item 3
Comment: Line 220 and 235: Reference is missing.
Response: Restored references to Table 3 and Table 5
Item 4
Comment: Table 3 is not present in the main text.
Response: The reference was one of the missing references mentioned in item 3, so fixing those also fixed the lack of reference to Table 3
Item 5
Comment: Please provide insights (in the main text or as a note in the table) on the “Amish Population Rank” in Table 3 and how it is assessed
Response: Added this. It’s just ranking the states from greatest to least in terms of Amish population.
Item 6
Comment: On line 226 you write: “The states with the greatest number of reports generally corresponded well to the states with the highest Amish population, with 4 of the top 5 in terms of number of reports also being in the top 5 in population.” Please review this quote. For example, Pennsylvania and Iowa have 4 and 3 reports, respectively, but the rank is 1 for Pennsylvania (84,100 population) and 9 for Iowa (9845 population). This means that a correlation between the number of reports and the population rank is not completely significant. I suggest reviewing the discussion of this data
Response: One of the other reviewers mentioned this as well and gave me some great recommendations for improving this section. I think following their suggestions for this section has addressed your comment as well.
Reviewer 4 Report
Good paper, but some issues can be improved before accepting:
- in Introduction some detailed comparison of official statistical data of accidents with horse driven and this AGNews has to be presented with detailed explanation why authors decide to use this kind of source. Also limitations of this kind of data source has to be highlighted
- line 192, some references missing, line 220 also, line 235 also, and check till the end
- because of very small sample I have doubt in statistical conclusions, also because Hisquare test is very low statistical test. So, I encourage authors to make some detailed analysis - in-depth studies of few accidents from the sample to show more detailed results
- to conclude my review - very interesting view of analysis but very small influence on road safety, because of framework border is limited only to few ''group'' of people. I expecting from authors to be more in details why they decide to do such kind of analysis. Authors put some reasons, but I want to see details, especially authors should give such conclusions if this kind of methodology is appropriate for using worldwide and for what cases.
-
Author Response
Dear Reviewer 4,
I appreciate the time and effort you have spent reviewing this article, An Assessment of Horse-Drawn Vehicle Incidents from U.S. News Media Reports within AgInjuryNews. Your feedback has led to substantial improvements in the article. I have carefully considered each item, and present my revisions in two formats. A brief response to each individual reviewer recommendation can be seen below. Additionally, changes made to the manuscript have been tracked and are highlighted in red. I hope the revisions have improved the quality of the manuscript, and welcome any further suggestions for improvement.
Item 1
Comment: in Introduction some detailed comparison of official statistical data of accidents with horse driven and this AGNews has to be presented with detailed explanation why authors decide to use this kind of source. Also limitations of this kind of data source has to be highlighted
Response: Added several sentences about the benefits and limitations of AgInjuryNews. As for official statistics, as far as I know there aren’t any, at least not at national level. The main two groups that seem to be conducting systematic surveillance of this population are the Pennsylvania DOT and the Purdue Extension in Indiana, which were already cited. They were reporting results mainly case counts, different ways of classifying incidents, and percentages.
Item 2
Comment: line 192, some references missing, line 220 also, line 235 also, and check till the end
Response: This was also noted by another reviewer and was corrected.
Item 3
Comment: because of very small sample I have doubt in statistical conclusions, also because Chisquare test is very low statistical test. So, I encourage authors to make some detailed analysis - in-depth studies of few accidents from the sample to show more detailed results
Response: Mentioned the limitation more clearly in the discussion.
As for in-depth studies of a few incidents, I added a few more details here and there. However, the paper had increased in length significantly after adding the additional data source recommended by reviewer two, so I did not feel as though there would be space for detailed case studies as well.
Item 4
Comment: to conclude my review - very interesting view of analysis but very small influence on road safety, because of framework border is limited only to few ''group'' of people. I expecting from authors to be more in details why they decide to do such kind of analysis. Authors put some reasons, but I want to see details, especially authors should give such conclusions if this kind of methodology is appropriate for using worldwide and for what cases
Response: The global angle is a good point, and one that I didn’t think of as someone who is primarily experiencing this issue in the context of rural roadways in the US. I’ve updated the text to make it clear that this could have global applications as well, and to emphasize that while the number of Anabaptist vehicles is low, this also affects the larger number of rural US residents who share the road with them.
Thank you again for your review!
Round 2
Reviewer 1 Report
The paper has improved slightly. But I still think it is of little interest to the scientific community. The results do not bring useful knowledge for improving road safety.
Author Response
Dear reviewer
I’d like to once again thank you for the time and effort you have spent reviewing this article, An Assessment of Horse-Drawn Vehicle Incidents from U.S. News Media Reports within AgInjuryNews. Your feedback has led to additional improvements in the article. I have carefully considered each item, and present my revisions in two formats. A brief response to each individual recommendation can be seen below. Additionally, changes made between the second and the current draft have been tracked and are highlighted in red. I hope the revisions have improved the quality of the manuscript, and welcome any further suggestions.
Item 1:
Comment:
The paper has improved slightly. But I still think it is of little interest to the scientific community. The results do not bring useful knowledge for improving road safety.
Response:
Again, it is fair enough to say that this will not impact every driver. However, this is an important issue for areas in the US, and possibly also in other countries, where there is a lot of interaction between motorized and horse-drawn vehicles.

Reviewer 2 Report
As noted earlier, the article is interesting and covers an important injury prevention topic. The author is to be applauded for taking on this difficult and challenging, but important subject area. I offer the following as encouragement to the author to continue to pursue improving the work.
As written, It is very confusingly organized. It would greatly improve the readability of this article if sections were introduced in both the methods and the results. Data source(s) should be incorporated into the results—It would be ok to include multiple data sources in the same table if footnotes were used to label the data source for each column heading.
There is a free website that is meant to support and guide authors publishing observational studies. I believe the author and article would benefit tremendously by examining and following some of the instructions and checklists that can be found at the free STROBE website(https://www.strobe-statement.org/) which is STrengthening the Reporting of OBservational studies in Epidemiology
The reader would benefit from the author using more precise language with regard to 3 items. Data tables confusingly appear to report the number of news reports (apparently culled for duplicative reports), number of crash incidents and individuals injured fatally and nonfatally, but these appear to be referred to sometimes in an interchangeable or unspecified format. It is likely that there are individuals involved in the incidents that are neither fatally or nonfatally injured, but if this is noted, I missed it. this is not noted. Which one is being referred to is not specified clearly. The logic behind some of the table data escapes me—table 3, if you have individuals injured or incidents, it could be informative to report these rather than number of news reports per 100,000 population. The final table has no number or label.
Minor comments: Table 1 (pages 5-6) has inconsistent use of capitalization in the listing of items. Capitalizing the first character of an item as is done in the first half of the table would be a preferred approach for the second portion also. .
Page 6, lines 240-241 could be improved by including percentages.
The word “Experienced” should be used in place of the word “suffered”.
Page 6, A quick search of the agnews data base did not turn up a codebook, but the website indicates that they report both incidents, victims (individuals) and individuals fatally injured. [i.e 4,167 Incidents, 5,507 Victims. 2,639 Fatalities]. How about crash individuals who are were at risk but did not get injured? Please use more precise language here to explain what the numbers are here—is this individual people or crashes (incidents)? This appears in the manuscript narrative on page 6. If it is 10 individuals involved in 8 crashes, this should be clarified. Or is this a typo (i.e. 8 incidents with 2 fatal and 8 nonfatal)? “Of the 8 on-road incidents, 2 were fatal and 8 were nonfatal”. In contrast, of the 13 off-road incidents, 9 were fatal and 2 were nonfatal. (are 2 missing fatal/nonfatal outcome or where they not injured?)
Page 6, line 247, being struck by a vehicle—is it known whether it was part of the fall or ejection and whether it was the buggy or the striking vehicle that ran over the injured party?
Table 1 and Table 2 report row percentages for the first 2 columns and column percentages for the last (total) column. It could be more informative if all three columns in both tables reported column percentages. Please use more precise table titles with the name of the data set included.
Table 3 on pages 7 and 8 lists 37 incidents. It would be good to use precise language in the titles and also the source of the data in the table title or a footnote.
For table 3 on pages 7 and 8, I might be missing something here, but if table 2 is counts of people is there a reason why the authors did not use population numerators from the table 2 incidents (fatal and nonfatal) with the population denominators in Table 3? Do the authors feel there is enough data to calculate number of injuries/deaths per 100,000 Amish population? Would this make more sense that number of articles?
Author Response
Dear Reviewer
I’d like to once again thank you for the time and effort you have spent reviewing this article, An Assessment of Horse-Drawn Vehicle Incidents from U.S. News Media Reports within AgInjuryNews. I can tell that you put a lot of time and thought into your feedback, and it has led to additional improvements in the article. I have carefully considered each item and present my revisions in two formats. A brief response to each individual recommendation can be seen below. Additionally, changes made between the second and the current draft have been tracked and are highlighted in red. I hope the revisions have improved the quality of the manuscript and welcome any further suggestions.
|
Comment |
Response |
|
It is very confusingly organized. It would greatly improve the readability of this article if sections were introduced in both the methods and the results. Data source(s) should be incorporated into the results—It would be ok to include multiple data sources in the same table if footnotes were used to label the data source for each column heading. |
I added some section headings and moved the section about the number of reports per state into the “comparison with other data” section.
Putting data from multiple sources into one table might work if we had 3 or more data sets we could compare across the same categories, but since each source handles things differently keeping them as separate tables makes more sense. |
|
There is a free website that is meant to support and guide authors publishing observational studies. I believe the author and article would benefit tremendously by examining and following some of the instructions and checklists that can be found at the free STROBE website(https://www.strobe-statement.org/) which is STrengthening the Reporting of OBservational studies in Epidemiology |
This is a great resource, and one that I had not heard of before. It will be useful for some of the other articles that I am working on too Thanks for sharing it! |
|
The reader would benefit from the author using more precise language with regard to 3 items. Data tables confusingly appear to report the number of news reports (apparently culled for duplicative reports), number of crash incidents and individuals injured fatally and nonfatally, but these appear to be referred to sometimes in an interchangeable or unspecified format. It is likely that there are individuals involved in the incidents that are neither fatally or nonfatally injured, but if this is noted, I missed it. this is not noted. Which one is being referred to is not specified clearly. The logic behind some of the table data escapes me—table 3, if you have individuals injured or incidents, it could be informative to report these rather than number of news reports per 100,000 population. The final table has no number or label. |
There are a few things mentioned here. Starting with the terminology, some of the tables are intentionally made at the victim level versus at the report or incident level. The victim-level tables focus on individual factors such as gender, age, role, and whether the person was fatally or non-fatally injured. The incident-level tables are focusing on the characteristics of the incident as a whole, such as involvement of a motor vehicle, rear-ending, etc. Where I switch to talking about the reports (1 report = 1 incident) is where I move away from talking about the incident itself to focusing on the database. I have changed some wording here and there to try to make this clearer, but the changes in terminology were deliberate.
As for people involved but who were not injured, it is really hit and miss whether that information was included or not, so they were not analyzed. There were only perhaps two or three examples of someone who was involved in the incident (as opposed to being a witness/bystander) being mentioned explicitly in the article, and one incident where I followed a link from the first article to an update article, which mentioned two more surviving siblings who were involved in the incident but not mentioned at all in the first article because they were not injured. It also seems to be inconsistent whether someone with minor injuries such as bruises or scrapes was counted as injured or not in the text. In practice what I did was to count the minor injuries as injuries and stuck to comparing fatal versus nonfatal since the presence of non-injured persons were not being reported reliably. I added a few sentences to the methods and discussion describing this difficulty.
As for Table 3, it has evolved from a table showing that there were generally more AgInjuryNews reports coming from states with higher Amish populations to reporting rates I went ahead and added columns for number of victims and rate per 100,000. It is important to remember, though, that both this number and the rate of incidents apply only to the AgInjuryNews Database. Since we know from the report from Purdue and the examination of Nexus Uni that AgInjuryNews is not capturing anywhere near all of the media reports, much less actual cases, I wanted to stay away from anything that would be mistaken as an estimate of real-world incidence rate. I added the rates for fatal and nonfatal injuries by state, but tried to make it clear in the writing that these are rates of reporting in the AgInjuryNews database only. |
|
Minor comments: Table 1 (pages 5-6) has inconsistent use of capitalization in the listing of items. Capitalizing the first character of an item as is done in the first half of the table would be a preferred approach for the second portion also. . |
Changed capitalization for consistency |
|
Page 6, lines 240-241 could be improved by including percentages.
|
Percentages added |
|
The word “Experienced” should be used in place of the word “suffered”
|
I originally had “experienced” but one my proofreaders wanted me to change it to “suffered”. I do not have strong opinions either way and am happy to go back to “experienced”. |
|
Page 6, A quick search of the agnews data base did not turn up a codebook, but the website indicates that they report both incidents, victims (individuals) and individuals fatally injured. [i.e 4,167 Incidents, 5,507 Victims. 2,639 Fatalities]. How about crash individuals who are were at risk but did not get injured? Please use more precise language here to explain what the numbers are here—is this individual people or crashes (incidents)? This appears in the manuscript narrative on page 6. If it is 10 individuals involved in 8 crashes, this should be clarified. Or is this a typo (i.e. 8 incidents with 2 fatal and 8 nonfatal)? “Of the 8 on-road incidents, 2 were fatal and 8 were nonfatal”. In contrast, of the 13 off-road incidents, 9 were fatal and 2 were nonfatal. (are 2 missing fatal/nonfatal outcome or where they not injured?) |
I also noticed this when I was adding the percentages on page 6. It was a typo. The numbers were supposed to be at the victim level. I fixed the typo and clarified wording.
As for people at risk for these injuries who were not injured, that is a really hard thing to get at for several reasons. Since AgInjuryNews relies on media reports (and even their coverage is imperfect) we are not able to get a reliable count of those injured. Getting a reliable count of Anabaptist residents is also a major difficulty mentioned in multiple articles. The US census does not capture data on religion, and the Amish are the only group that (at least publicly) periodically releases counts of church membership. Being able to establish a rate would be great, but neither the numerator or the denominator is reliable in this case. |
|
Page 6, line 247, being struck by a vehicle—is it known whether it was part of the fall or ejection and whether it was the buggy or the striking vehicle that ran over the injured party? |
There are some cases where it is known whether it was a runover/strike without a fall (example: child running out in front of farm equipment), a fall/ejection with out a runover/strike (example: buggy driver thrown clear of a crash with a motor vehicle) and in some it is clear that it was both (example: buggy passenger fell/was ejected then run over by a motor vehicle). In other cases it is not clear, and in many cases the distinction between a fall vs ejection or strike vs runover are not clear, which is why these variables are grouped. I added a description with examples to help clarify. |
|
Table 1 and Table 2 report row percentages for the first 2 columns and column percentages for the last (total) column. It could be more informative if all three columns in both tables reported column percentages. Please use more precise table titles with the name of the data set included |
Changed Table 1 and Table 2 to all column percentages
Both of these tables already had “from the AgInjuryNews Dataset” in the titles, and are not comparing to another source. |
|
Table 3 on pages 7 and 8 lists 37 incidents. It would be good to use precise language in the titles and also the source of the data in the table title or a footnote. |
The data sources for Table 3 were already in the title and text, but I tried to clarify further. |
|
For table 3 on pages 7 and 8, I might be missing something here, but if table 2 is counts of people is there a reason why the authors did not use population numerators from the table 2 incidents (fatal and nonfatal) with the population denominators in Table 3? Do the authors feel there is enough data to calculate number of injuries/deaths per 100,000 Amish population? Would this make more sense that number of articles? |
It is very clear that AgInjuryNews is not capturing all media articles available, much less making anywhere near a complete count all incidents occurring at the population level. Determining the population itself is also somewhat difficult because the Amish are the only Anabaptist group that periodically releases statistics on their number of members. Since they are by far the largest group you can make an argument for using their population as an overall estimate. Given the known lack of coverage of AgInjuryNews though, it’s not nearly complete enough to claim that the reporting rate is anywhere near the incidence rate in the population. The studies I read to prepare for this article also often cited the difficulty in accurately establishing the population, much less reliable injury and fatality rates. I’ve added a couple of sentences to try to clarify this point. |

Reviewer 3 Report
Thank you for the answers.
Author Response
Dear Reviewer
I’d like to once again thank you for the time and effort you have spent reviewing this article, An Assessment of Horse-Drawn Vehicle Incidents from U.S. News Media Reports within AgInjuryNews. While I did not recieve any specific feedback from you, changes made between the second and the current draft based on the feedback of the other reviewers have been tracked and are highlighted in red. I hope the revisions have improved the quality of the manuscript, and welcome any further suggestions.

Reviewer 4 Report
Authors response to all my review, so I am recommending publishing.
Author Response

(The authors gave the same response as above.)

Round 3
Reviewer 1 Report
I still believe that the paper is not of interest to the scientific community. Both for the topic dealt with and for the analysis methodology.
Author Response
I still believe that the paper is not of interest to the scientific community. Both for the topic dealt with and for the analysis methodology
Again, while the interest isn't going to be universal, this is still a significant issue for communities affected and in other areas of the world where other types of horse-drawn vehicles are common.
Reviewer 2 Report
I do not have further suggestions over what I originally provided except that the AgNews is not a surveillance system of injury. Attempting to analyze it as such may be the source of the bias--see below:
1) The AgNews contains a footnote that says "*There is no official national dataset of all agricultural injuries and fatalities, however news reports can be helpful in understanding the scope of the problem."
2) Use of the term "bias" has negative connotations when the author could get the point across using less loaded terms. Given the above note on the website that contains the data used in this study, something that notes that news reports capture more serious injury and fatality and do not necessarily report on less serious injury seems appropriate for the body of the paper and for the conclusions. The AgNews never set out to be a surveillance system so to call them biased because they do not meet the criteria of a surveillance system of all injury is too negative and this point should be made in the paper and the conclusions. Use of the term "emphasis on more serious injury/death over less serious injury", could be more appropriate than saying a news outlet is "biased". The term bias implies unreasoned and unfair, not just unbalanced.
Author Response
1) The AgNews contains a footnote that says "*There is no official national dataset of all agricultural injuries and fatalities, however news reports can be helpful in understanding the scope of the problem."
I wasn't sure if I was being asked to add an identical footnote specifically, but I went through and clarified at several points in the article that this is a database of media reports and not an attempt at comprehensive surveillance.
2) Use of the term "bias" has negative connotations when the author could get the point across using less loaded terms. Given the above note on the website that contains the data used in this study, something that notes that news reports capture more serious injury and fatality and do not necessarily report on less serious injury seems appropriate for the body of the paper and for the conclusions. The AgNews never set out to be a surveillance system so to call them biased because they do not meet the criteria of a surveillance system of all injury is too negative and this point should be made in the paper and the conclusions. Use of the term "emphasis on more serious injury/death over less serious injury", could be more appropriate than saying a news outlet is "biased". The term bias implies unreasoned and unfair, not just unbalanced.
Replaced all instances of the word "bias" with alternate wordings suited to each individual instance. Also added emphasis in several places that news media reports are different from comprehensive surveillance.